# Diff-eRank: A Novel Rank-Based Metric for Evaluating Large Language Models

**Lai Wei**[1,*]   **Zhiquan Tan**[2,*]   **Chenghai Li**[4]   **Jindong Wang**[3]   **Weiran Huang**[1,†]

[1] MIFA Lab, Qing Yuan Research Institute, SEIEE, Shanghai Jiao Tong University
[2] Department of Mathematical Sciences, Tsinghua University
[3] William & Mary    [4] Independent

## Abstract

Large Language Models (LLMs) have transformed natural language processing and extended their powerful capabilities to multi-modal domains. As LLMs continue to advance, it is crucial to develop diverse and appropriate metrics for their evaluation. In this paper, we introduce a novel rank-based metric, *Diff-eRank*, grounded in information theory and geometry principles. Diff-eRank assesses LLMs by analyzing their hidden representations, providing a quantitative measure of how efficiently they eliminate redundant information during training. We demonstrate the applicability of Diff-eRank in both single-modal (e.g., language) and multi-modal settings. For language models, our results show that Diff-eRank increases with model size and correlates well with conventional metrics such as loss and accuracy. In the multi-modal context, we propose an alignment evaluation method based on the eRank, and verify that contemporary multi-modal LLMs exhibit strong alignment performance based on our method. Our code is publicly available at https://github.com/waltonfuture/Diff-eRank.

## 1   Introduction

Large Language Models (LLMs) such as GPT [4, 23], Chinchilla [17], and PaLM [7], have gained considerable attention for their outstanding performance in various natural language processing tasks. LLMs have expanded from single-modal models to multi-modal models, including MiniGPT-4 [49] and LLaVA [20], which have achieved remarkable results in various application scenarios. Pre-trained LLMs rely on large networks, computational power, and massive amounts of data, aiming for greater generalization capabilities.

LLMs understand the world knowledge through training on huge amounts of data. One famous belief [34] of how LLMs work is that larger models can find more shared hidden structures in data samples by eliminating redundant information through training. In particular, in the early phase of training, following random initialization, the representations derived from the training data tend to be somewhat chaotic. As training progresses, these representations become increasingly structured, and the model discards extraneous information from the training data, which resembles a process similar to "noise reduction". This perspective motivates us that LLM could be evaluated by characterizing the "noise reduction" process.

However, defining and quantifying the degree of "noise reduction" remains a significant challenge. To address this, we hypothesize that a reasonable metric should 1) reflect the geometric characteristics of the data such as the dimensionality of its representations, and 2) be rooted in information theory.

---

*Lai Wei (waltonfuture@sjtu.edu.cn) and Zhiquan Tan (tanzq21@mails.tsinghua.edu.cn) contributed equally.
†Correspondence to Weiran Huang (weiran.huang@outlook.com).

In this paper, we introduce *Diff-eRank* (difference between effective ranks), an information-theoretic metric that fulfills both criteria, providing a measure for quantifying "noise reduction" in LLMs. In particular, we consider the *effective rank* (eRank) of the representations extracted by an LLM from a dataset to measure the uncertainty, based on concepts from (quantum) information theory [30]. Through the removal of redundant information, eRank decreases, indicating the representations become more structured and compact. Thus, the reduction of representations' eRank can signify the degree of "noise reduction". Therefore, we can evaluate a well-trained LLM via the eRank reduction of the model representations *from its untrained status*. We remark that different from conventional metrics like loss, which are derived from the predictions of LLMs, the proposed Diff-eRank focuses on the model representations. Our approach offers a novel perspective of model assessment, independent of prediction-based metrics, and can provide new insights into the understanding of LLM's behavior.

To verify the effectiveness of our approach, we conduct experiments on the contexts of both uni-modal LLMs and multi-modal LLMs. In particular, for uni-modal LLMs, we compute Diff-eRanks for models within the OPT family [45] across various datasets. Intriguingly, we observe that Diff-eRank increases as the model scales, suggesting that larger models exhibit a stronger noise reduction ability. Moreover, Diff-eRank has a consistent trend when compared with (reduced) cross-entropy loss and benchmark accuracy, highlighting its potential as an effective and easy-to-use evaluation metric. For multi-modal (vision-language) LLMs, visual and language information is usually encoded separately by two independent encoders and aligned through a connecting layer. Therefore, evaluating the quality of modality alignment in multi-modal LLMs is crucial. Building on insights from uni-modal LLMs, we can assess modality alignment by examining the matching degree of eRanks between representations from different modalities. Additionally, this approach yields interesting observations within the context of such multi-modal architectures.

Our contribution can be summarized as follows:

- We propose a rank-based metric, Diff-eRank, for evaluating LLMs, where Diff-eRank reflects the "noise reduction" ability of pre-trained language models. Diff-eRank focuses on the model representations, different from conventional metrics such as loss and benchmark accuracy.

- We validate the effectiveness of Diff-eRank by observing its correlation with the trends in loss and downstream task accuracy as the model scales up.

- We also propose eRank-based modality alignment metrics for multi-modal LLMs, and verify that contemporary multi-modal LLMs exhibit strong alignment performance via our metrics.

## 2 Related Works

**Evaluation of Large Language Models.** Evaluation of LLMs is a fast-evolving field across various tasks, datasets, and benchmarks [5, 33, 36, 48]. Precise evaluations are important for the enhancement of language models' performance and reliability. Conventional metrics such as accuracy, F1 [29], BLEU [24] and ROUGE [18] estimate between the annotated label and the prediction generated by the language model in different downstream tasks. Other metrics like perplexity and cross-entropy loss are independent of annotated labels and can be computed solely based on the input texts. However, these metrics focus on "extrinsic" evaluation, assessing performance based on the predictions of LLMs. We propose Diff-eRank for "intrinsic" evaluation based on the input data's hidden representations of LLMs, concentrating on their "noise reduction" capabilities.

**Information Theory for Understanding Deep Learning.** Information theory has been used to gain significant insights into understanding neural networks. For example, the information bottleneck [37, 38] is instrumental in explaining supervised learning. Recently, researchers have also utilized information theory to understand and improve (vision) semi and self-supervised learning [32, 35, 46, 47]. Notably, Zhang et al. [46] find the closed-form connection of matrix entropy and effective rank when the matrix is positive semi-definite. As for language models, prior works [16, 25, 41] also used information theory to analyze hidden representations by training probes on specific downstream tasks to estimate the information contained in the pre-trained language model. Several other works explore the lossless compression of LLMs with arithmetic coding [10, 40] based on information theory. In this paper, we take a further step toward evaluating LLMs through the proposed Diff-eRank rooted in information theory, which represents a complementary perspective to these prior studies.

# 3 The Proposed Metric for Evaluating LLMs

In this section, we will introduce a rank-based metric called *Diff-eRank* for evaluating LLMs. The proposed metric is based on the representations obtained by an LLM, fundamentally diverging from conventional metrics like loss, which are based on the model's predictions.

When processing a sequence of tokens, an LLM will generate a representation (i.e., the hidden states before the last classification head) for each token within the sequence. These high-dimensional representations are usually used to capture the semantic and syntactic information of the sentences. This inspires us to consider evaluating LLMs by analyzing these representations. In particular, we study the characteristics of these representations by examining their ranks through both the geometric and information-theoretic perspective. On the one hand, studying the rank of these representations allows us to measure the extent of linear independence among them, which corresponds to the effective dimensions in the representation space (i.e., the geometric structure). On the other hand, the rank is also related to the amount of information contained in these representations, while a lower rank indicates that the information has been structured or compressed. Therefore, we consider to leverage the rank of data representations encoded by LLMs for model evaluation.

However, the size of data representation matrix varies with the sample size, making it less suitable for consistent analysis. Therefore, instead of directly computing the rank of the data representations, we use the rank of their *covariance matrix*, which has a fixed size and also contains all the essential information. To see this, let $\mathcal{S} = \{\mathbf{z}_1, \mathbf{z}_2, \ldots, \mathbf{z}_N\}$ denote the set of data representations, and $\bar{\mathbf{z}}$ be the mean representation. The rank of data representation matrix can be re-formulated as

$$\mathrm{rank}([\mathbf{z}_1 - \bar{\mathbf{z}}, \cdots, \mathbf{z}_N - \bar{\mathbf{z}}]) = \mathrm{rank}\left(\frac{1}{N}[\mathbf{z}_1 - \bar{\mathbf{z}}, \cdots, \mathbf{z}_N - \bar{\mathbf{z}}][\mathbf{z}_1 - \bar{\mathbf{z}}, \cdots, \mathbf{z}_N - \bar{\mathbf{z}}]^\top\right)$$

$$= \mathrm{rank}\left(\frac{1}{N}\sum_{i=1}^{N}(\mathbf{z}_i - \bar{\mathbf{z}})(\mathbf{z}_i - \bar{\mathbf{z}})^\top\right),$$

where the last term is exactly the rank of covariance matrix. We remark that the above rank also equals to the dimension of the affine subspace spanned by $\mathcal{S} \cup \{\bar{\mathbf{z}}\}$.

The formal construction of covariance matrix is shown as follows. For ease of analysis, each $\mathbf{z}_i - \bar{\mathbf{z}}$ is being normalized.

**Definition 3.1** (Construction of Covariance Matrix). Given a set of representations $\mathcal{S} = \{\mathbf{z}_i \in \mathbb{R}^d \mid i = 1, 2, \ldots, N\}$, the covariance matrix $\mathbf{\Sigma}_\mathcal{S}$ is constructed as

$$\mathbf{\Sigma}_\mathcal{S} = \frac{1}{N}\sum_{i=1}^{N}\left(\frac{\mathbf{z}_i - \bar{\mathbf{z}}}{\|\mathbf{z}_i - \bar{\mathbf{z}}\|}\right)\left(\frac{\mathbf{z}_i - \bar{\mathbf{z}}}{\|\mathbf{z}_i - \bar{\mathbf{z}}\|}\right)^\top,$$

where $\bar{\mathbf{z}} = \sum_{i=1}^{N}\mathbf{z}_i/N$ is the mean representation and notation $\|\cdot\|$ represents $\ell_2$ norm.

Since rank is highly sensitive to outliers [27], we instead use its "continuous" counterpart, the *effective rank* (eRank), when applied to the covariance matrix, defined as below.

**Definition 3.2** (eRank [27]). The effective rank of any non-zero matrix $\mathbf{A} \in \mathbb{R}^{d \times N}$ is defined as

$$\mathrm{eRank}(\mathbf{A}) = \exp\left(-\sum_{i=1}^{Q}\frac{\sigma_i}{\sum_{i=1}^{Q}\sigma_i}\log\frac{\sigma_i}{\sum_{i=1}^{Q}\sigma_i}\right),$$

where $Q = \min\{N, d\}$ and $\sigma_1, \sigma_2, \ldots, \sigma_Q$ are the singular values of matrix $\mathbf{A}$.

We remark that the above eRank is closely related to the matrix entropy (i.e., Von Neumann entropy for matrices [42]), which is defined in Definition 3.3. In fact, Zhang et al. [46] point out that, for a covariance matrix of *normalized* vectors, $\mathrm{eRank}(\mathbf{\Sigma}_\mathcal{S})$ is the same as $\exp(\mathrm{H}(\mathbf{\Sigma}_\mathcal{S}))$.

**Definition 3.3** (Matrix Entropy). Given a positive semi-definite matrix $\mathbf{K} \in \mathbb{R}^{d \times d}$ satisfying $\mathrm{tr}(\mathbf{K}) = 1$, the matrix entropy of matrix $\mathbf{K}$ is defined as

$$\mathrm{H}(\mathbf{K}) = -\mathrm{tr}(\mathbf{K}\log\mathbf{K}).$$

It is equivalent to the Shannon entropy [31] over the spectrum, i.e.,

$$\mathrm{H}(\mathbf{K}) = -\sum_{i=1}^{d} \lambda_i \log \lambda_i,$$

where $\lambda_1, \lambda_2, \ldots, \lambda_d$ are the eigenvalues of matrix $\mathbf{K}$.

Note that eRank of the covariance matrix is commonly interpreted as a measure of the "degree of freedom" that the sentence contains in a geometric sense, one may wonder whether there is a more "information-theoretic" explanation for it. Interestingly, under the terminology of quantum information theory [44], if we regard the representation of each token as a *state* in a quantum system, the construction given by Definition 3.1 is a standard process of constructing a *density matrix*. From the quantum noiseless coding theorem [30], the entropy of a density matrix $\mathrm{H}(\mathbf{\Sigma}_{\mathcal{S}})$ represents the average number of qubits required to encode the states. Therefore, $\exp\left(\mathrm{H}(\mathbf{\Sigma}_{\mathcal{S}})\right)$ can be viewed as a measure of randomness for a sentence through the quantum information theory.

As eRank measures the amount of uncertainty in a system, we can now define Diff-eRank to measure the degree of "noise reduction" for an LLM.

**Definition 3.4** (Diff-eRank). Given a sentence $x$, an untrained language model $M_0$, and a compute-optimal [17] trained language model $M_1$, we obtain two sets of representations, $M_0(x)$ and $M_1(x)$, by processing each token of $x$ through the respective models. Then the rank difference (i.e., Diff-eRank) between these two models based on sentence $x$ is defined as follows:

$$\Delta\,\mathrm{eRank}(x, M_0, M_1) = \mathrm{eRank}\left(\mathbf{\Sigma}_{M_0(x)}\right) - \mathrm{eRank}\left(\mathbf{\Sigma}_{M_1(x)}\right),$$

where $\mathbf{\Sigma}_{M_i(x)}$ is the covariance matrix of model $M_i$'s representations on sentence $x$ for $i \in \{0, 1\}$.

Upon completing training, the model's data representations shift from being random to more structured, enabling it to effectively capture patterns and structures from the data. In the above definition, the effective ranks $\mathrm{eRank}(\mathbf{\Sigma}_{M_0(x)})$ and $\mathrm{eRank}(\mathbf{\Sigma}_{M_1(x)})$ quantify the uncertainty in the representations of the untrained and trained models, respectively. Thus, Diff-eRank $\Delta\,\mathrm{eRank}(x, M_0, M_1)$ measures how much uncertainty the model has reduced as a result of training.

The above definition applies to a single sentence but can be extended to a dataset consisting of multiple sentences. Specifically, Diff-eRank for the entire dataset can be defined as the average Diff-eRank across all sentences, formulated as follows.

**Definition 3.5** (Diff-eRank of a Dataset). Given a dataset $\mathcal{D}$ consisting of sentences $x_1, \ldots, x_n$, an untrained language model $M_0$, and a compute-optimal [17] trained language model $M_1$, Diff-eRank of dataset $\mathcal{D}$ is defined as

$$\Delta\,\mathrm{eRank}(\mathcal{D}, M_0, M_1) = \exp\left(\frac{\sum_{i=1}^{n} \mathrm{H}\left(\mathbf{\Sigma}_{M_0(x_i)}\right)}{n}\right) - \exp\left(\frac{\sum_{i=1}^{n} \mathrm{H}\left(\mathbf{\Sigma}_{M_1(x_i)}\right)}{n}\right).$$

In summary, Diff-eRank reflects the dimension reduction of the space spanned by data representations. It can be viewed as a measure of removing redundant information in the data for a compute-optimal language model. A higher Diff-eRank indicates more organized and structured internal representations of the model, therefore reflecting the model's increasing effectiveness in capturing patterns and regularities in the data.

## 4 Evaluations of Large Language Models

We start with evaluating different sizes of language models via Diff-eRank in Section 4.2. We find that Diff-eRank increases as the model scales up on various datasets. Additionally, we extend the application of eRank to multi-modalities beyond the language domain in Section 5.

### 4.1 Experimental Settings

#### 4.1.1 Model Choice

We experiment by using popular transformer-based language models from OPT [45] family, ranging from 125 million to 13 billion parameters. Such diversity in OPT's model size allows for a

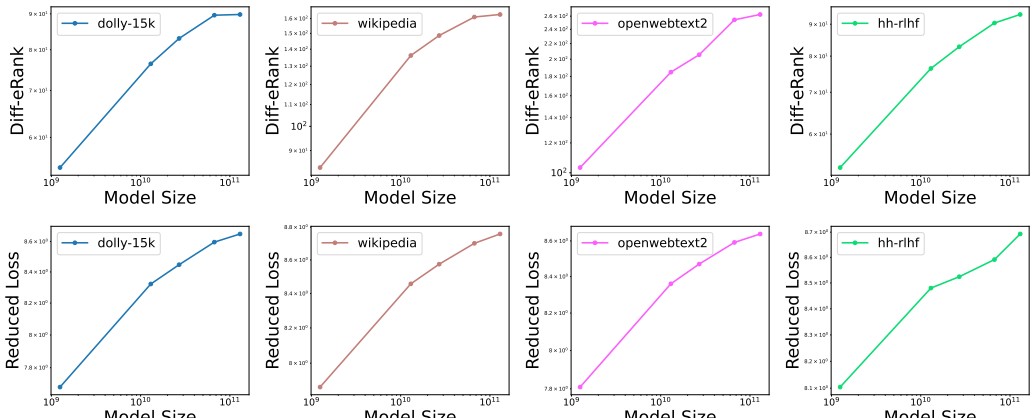

Figure 1: Comparison of Diff-eRank and reduced loss when model scales up across various datasets. Both Diff-eRank and reduced loss show an upward trend when the model scales up.

comprehensive analysis across different scales of pre-trained language models in our experimental setting. We refer the reader to Appendix A for additional implementation details about the selection of language datasets.

### 4.1.2 Metric for Comparison

Given a text sequence $U = [u_1, \ldots, u_T]$, the cross-entropy loss of a language model $M$ can be defined as

$$L(U, M) = -\frac{1}{T} \sum_{i=1}^{T} \log P(u_i | u_1, \ldots, u_{i-1}).$$

The cross-entropy loss is a canonical quantity in Shannon information theory, based on the model's predictions. As we study the rank difference between untrained model $M_0$ and compute-optimal trained model $M_1$ based on representation, we adopt the difference in loss for comparison, correspondingly. Therefore, we can similarly define reduced (cross-entropy) loss as

$$\Delta L(U, M_0, M_1) = L(U, M_0) - L(U, M_1).$$

As the training progresses, the LLM gets better predictions on the input data, leading to an increase in reduced loss. Therefore, reduced loss can also be seen as a useful evaluation metric for LLMs, and we use it for comparison with Diff-eRank in our following experiments.

### 4.2 The Trend of Diff-eRank with Model Size

To substantiate Diff-eRank as a viable metric for evaluation, we evaluate the series of OPT [45] models over different and diverse datasets using Diff-eRank and (reduced) loss for comparison. Specifically, we consider including pre-training datasets such as Wikipedia [14] and openwebtext2 [15], instruction dataset dolly-15k [8], and preference dataset hh-rlhf [2] for the diversity of their usage.

Figure 1 presents that Diff-eRank and reduced loss both increase progressively as the model scales up. The increase in reduced loss (equals to a decrease in cross-entropy loss) can be interpreted as larger models providing closer predictions to the actual values so that they can better capture the underlying patterns and relationships within the data. As for the increase in Diff-eRank based on hidden representations, it suggests that the redundant dimensions of the data can be effectively reduced in the larger models, thereby resulting in stronger "noise reduction" abilities and larger Diff-eRanks. Overall, the strong correlation between Diff-eRank and (reduced) loss indicates that Diff-eRank can provide a novel and inspirational evaluation for LLMs through the lens of "noise reduction" in dimension spaces. We summarize detailed results tables in Appendix B.

### 4.3 Relationship among Benchmark Metrics

Based on the exploration in the earlier section, a natural question arises: does Diff-eRank relate to the downstream task accuracy of large language models? To address this question, we integrate accuracy

Table 1: Comparison of benchmark metrics on openbookqa [22] and piqa [3]. ACC denotes benchmark accuracy and $\Delta L$ indicates reduced loss. The results indicate that larger Diff-eRank values generally correspond to higher model performance.

| BENCHMARKS | INDICATORS | OPT MODELS SIZE | | | | |
|---|---|---|---|---|---|---|
| | | 125M | 1.3B | 2.7B | 6.7B | 13B |
| OPENBOOKQA | ACC | 0.276 | 0.332 | 0.370 | 0.360 | **0.366** |
| | $\Delta L$ | 5.734 | 6.138 | 6.204 | **6.258** | 6.236 |
| | DIFF-ERANK | 1.410 | 2.140 | 2.338 | 2.280 | **3.032** |
| PIQA | ACC | 0.619 | 0.714 | 0.733 | 0.756 | **0.767** |
| | $\Delta L$ | 6.472 | 6.928 | 6.999 | **7.077** | 7.068 |
| | DIFF-ERANK | 4.647 | 6.294 | 6.774 | 6.950 | **7.267** |

as a comparative metric in addition to Diff-eRank and reduced loss in our evaluations on benchmark datasets. We use the evaluation set of openbookqa [22] and piqa [3] by combining the question and correct answer of each piece of data as inputs.

The results presented in Table 1 illustrate a similar relationship among Diff-eRank, reduced loss, and downstream task accuracy. All of these three metrics increase when model size increases. Although occasional outliers are observed in the upward trends of these indicators, we think this is normal and does not affect the overall trend. Therefore, it can be concluded that Diff-eRank generally correlates with the trend of loss and accuracy, particularly as the model size scales within the same model family. An increase in Diff-eRank (i.e., a higher denoising ability of the model) corresponds to enhanced model performance (i.e., higher reduced loss and higher accuracy), which shows great potential in the evaluation of LLMs.

# 5 Evaluations of Multi-Modal Large Language Models

After verifying that Diff-eRank can indeed reflect the LLMs' intrinsic ability in the previous sections, our study extends to the evaluation of Multi-modal Large Language Models (MLLMs) [1, 20, 43, 49]. We define new metrics based on the eRank to evaluate the *modality alignment*.

## 5.1 Experimental Settings

For our multi-modal experiments, we select two advanced and open-sourced MLLMs as shown in Table 5 in the appendix: LLaVA-1.5 [19] and MiniGPT-v2 [6]. Both the two MLLMs utilize a simple connector for aligning the vision encoder with the LLM, providing a streamlined approach to multi-modal learning. We conduct the experiments on two high-quality multi-modal instruction datasets: detail_23k [20] and cc_sbu_align [49]. Each piece of data in these datasets contains a triplet of image, instruction, and response. We concatenate the instruction and response of each triplet as the textual input in our experiments.

## 5.2 Empirical Observations

Most of the MLLMs typically employ a projector mechanism (usually linear layer or MLP), which aligns image representations from a vision encoder (usually ViT [12]) with LLM's language representations. Our experiments include analyzing the effective rank of representation of images post vision encoder ($\text{eRank}_1$) and post connector ($\text{eRank}_2$), as well as the representation output by the LLM for individual images ($\text{eRank}_3$), text ($\text{eRank}_4$), and image-text pairs ($\text{eRank}_5$), as shown in Figure 2. To measure the "modality alignment" of MLLMs, we introduce two distinct metrics based on eRank:

$$\text{Image Reduction Ratio} = \frac{\text{eRank}_1 - \text{eRank}_2}{\text{eRank}_1},$$

and

$$\text{Image-Text Alignment} = \frac{\text{avg}(\text{eRank}_3, \text{eRank}_4, \text{eRank}_5)}{\text{max}(\text{eRank}_3, \text{eRank}_4, \text{eRank}_5)}.$$

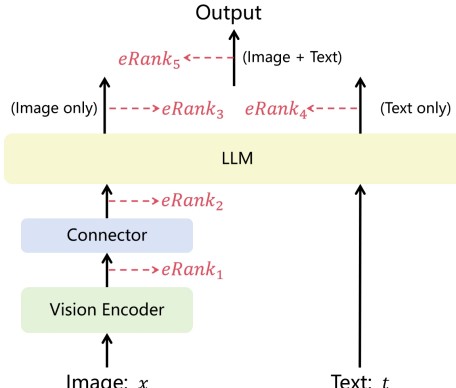

Output

$eRank_5$ ⟵--- (Image + Text)

(Image only) ---⟶ $eRank_3$    $eRank_4$ ⟵--- (Text only)

LLM

⟵---⟶ $eRank_2$

Connector

⟵---⟶ $eRank_1$

Vision Encoder

Image: $x$                Text: $t$

Figure 2: Illustration of the eRank measurement in the MLLM framework. The evaluation encompasses the effective rank of image representations after the vision encoder ($eRank_1$), post-connector representations ($eRank_2$), as well as the output representations generated by the LLM including individual images ($eRank_3$), textual data ($eRank_4$), and the combined image-text pairs ($eRank_5$).

Table 2: Multi-modal LLMs' results. "Image Reduction Ratio" and "Image-Text Alignment" measure the degree of "modality alignment" based on eRank.

| EFFECTIVE RANK | LLAVA-1.5 | | MINIGPT-V2 | |
|---|---|---|---|---|
| | DETAIL_23K | CC_SBU_ALIGN | DETAIL_23K | CC_SBU_ALIGN |
| $eRank_1$ | 18.34 | 9.00 | 90.59 | 74.79 |
| $eRank_2$ | 11.28 | 5.20 | 55.70 | 46.15 |
| $eRank_3$ | 45.62 | 28.47 | 58.50 | 48.68 |
| $eRank_4$ | 74.21 | 59.00 | 63.63 | 52.68 |
| $eRank_5$ | 76.34 | 47.63 | 108.53 | 93.29 |
| IMAGE REDUCTION RATIO (↑) | 0.3850 | 0.4222 | 0.3851 | 0.3829 |
| IMAGE-TEXT ALIGNMENT (↑) | 0.8566 | 0.7618 | 0.7084 | 0.6955 |

Table 3: Results of the image operation by clockwise rotating.

| EFFECTIVE RANK | LLAVA-1.5 ON DETAIL_23K | |
|---|---|---|
| | BASE | ROTATE IMAGE CLOCKWISE |
| $eRank_1$ | 18.34 | 19.20 (↑) |
| $eRank_2$ | 11.28 | 12.31 (↑) |
| $eRank_3$ | 45.62 | 46.54 (↑) |
| $eRank_4$ | 74.21 | 74.21 (-) |
| $eRank_5$ | 76.34 | 77.69 (↑) |
| IMAGE REDUCTION RATIO | 0.3850 | 0.3588 (↓) |
| IMAGE-TEXT ALIGNMENT | 0.8566 | 0.8514 (↓) |

On the one hand, the "*Image Reduction Ratio*" metric is formulated to quantify the reduction in effective rank from the vision encoder output ($eRank_1$) to the post-connector stage ($eRank_2$). Note that normalization is necessary here for a fair comparison because the vision encoder and connector are entirely different networks. This metric evaluates the connector network's efficiency in condensing and refining visual information during image-text alignment training. On the other hand, the "*Image-Text Alignment*" metric is designed to evaluate the closeness among the effective rank of representations post LLM processing, considering individual images ($eRank_3$), text ($eRank_4$), and image-text pairs ($eRank_5$) as inputs. In particular, the absolute eRank can be seen as the amount of absolute uncertainty or randomness. The mentioned three eRanks show how much the model integrates and represents each modality. If these three eRanks from different modalities are close to each other, it means that they align well from the perspective of information theory. Thus, this metric reflects the degree of closeness (i.e., alignment) among different modalities. A higher alignment score indicates a more proficient alignment between image and text modalities for MLLMs.

Results in Table 2 exhibit the performance of two MLLMs, LLaVA-1.5 [19] and MiniGPT-v2 [6], across different datasets (detail_23k [20] and cc_sbu_align [49]). Both models align well as they all have a relatively high alignment score.

In particular, comparing the two models, LLaVA-1.5 and MiniGPT-v2 both exhibit similar "Image Reduction Ratio" scores, indicating efficient condensation of visual information. Additionally, LLaVA-1.5 outperforms MiniGPT-v2 in "Image-Text Alignment", suggesting a closer integration between visual and textual modalities. This finding is also consistent with their performance, as LLaVA-1.5 surpasses MiniGPT-v2 in most of benchmarks [9]. We leave exploring a more comprehensive evaluation for multi-modal models via effective rank as future work.

To further investigate the role of each component in MLLM, we conduct additional experiments to calculate the eRank after rotating the images clockwise. We summarize the results in Table 3. As the rotation of images introduces new semantic information into the model, by noticing all the image-related quantities ($eRank_i$ ($i \neq 4$)) all *increase* from the base model when performing rotation, this semantic influence can propagate through the model. Therefore, we suggest that the multi-modal model (including the connector and the language model) can indeed perceive subtle semantic variations in images, especially the position information. In addition, the "Image Reduction Ratio" score and "Image-Text Alignment" score both *decrease* after conducting image rotation, suggesting that the connector performs less effectively in condensing visual information, and the rotated images

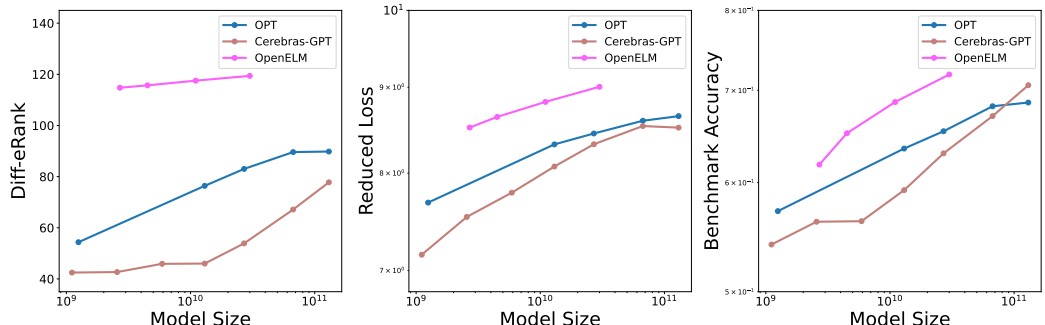

Figure 3: Comparing Diff-eRank with reduced loss and benchmark accuracy across different model families, including OPT [45], Cerebras-GPT [11], and OpenELM [21].

are less well-aligned with the corresponding text. This is primarily because the rotation alters the spatial relationships within the image, possibly making it more challenging for the model to maintain the coherence between visual and textual information. Overall, this experiment indicates that subtle changes in the vision encoder's understanding of images can be effectively conveyed to the LLM part and affect the MLLM's modality alignment. It demonstrates the validity of such a popular multi-modal architecture.

In conclusion, these rank-based approaches enable a thorough understanding of how well the multi-modal models align different modalities of data and how the models process and integrate different forms of input data.

## 6  Ablation Study

To better confirm the rationality of our algorithm and experimental design, we further conduct a series of ablation studies.

### 6.1  Different Model Families

Besides observing Diff-eRank on the OPT family, we also conduct experiments on Cerebras-GPT [11] family and OpenELM [21] family. LLMs in these three families are all pre-trained well on public data and range in various sizes. To demonstrate that Diff-eRank is not dependent on specific datasets, we choose not to use benchmark datasets but instead select a general dataset. In particular, we adopt the dolly-15k [8] dataset to compute Diff-eRank along with reduced loss, and we calculate the average benchmark accuracy of winogrande [28] and piqa [3] for these three LLM families. The empirical findings in Figure 3 substantiate the increase of Diff-eRank within these LLM families as the models scale up, which correlates with the trend of reduced loss and benchmark accuracy. This observation shows the potential of Diff-eRank as an insightful metric for the evaluation of different model families.

### 6.2  Algorithm Design

In this section, we choose other types of algorithms for designing Diff-eRank between untrained model $M_0$ and trained model $M_1$. The goal is to validate that the increasing type relation is robust to the algorithm we used.

We denote our standard computation of effective rank on a dataset $\mathcal{D}$ (Definition 3.5) as "Algorithm (a)", which calculates the effective rank based on the *average matrix entropy*. In addition, we also consider the operation of calculating the *average effective rank* on a dataset $\mathcal{D}$, denoted by "Algorithm (b)". Specifically, for an LLM $M$, the effective rank on a dataset $\mathcal{D}$ of Algorithm (b) is defined as

$$\mathrm{eRank}^{(b)}(\mathcal{D}, M) = \frac{\sum_{x \in \mathcal{D}} \exp(\mathrm{H}(\mathbf{\Sigma}_{M(x)}))}{|\mathcal{D}|} = \frac{\sum_{x \in \mathcal{D}} \mathrm{eRank}(\mathbf{\Sigma}_{M(x)})}{|\mathcal{D}|}.$$

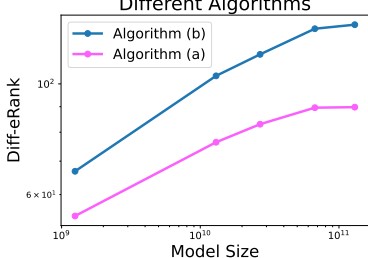

Figure 4: Different designs for Diff-eRank.

Table 4: Diff-eRank on different layers of OPT models. Only the Diff-eRank on the last layer indicates an increasing trend.

| OPT MODELS | 125M | 1.3B | 2.7B | 6.7B | 13B |
|---|---|---|---|---|---|
| FIRST LAYER | 73.07 | 73.03 | 66.93 | 49.24 | 41.83 |
| MIDDLE LAYER | 87.75 | 51.98 | 56.16 | 66.63 | 73.88 |
| LAST LAYER ($\uparrow$) | 54.35 | 76.39 | 83.02 | 89.60 | 89.81 |

Therefore, Diff-eRank between untrained model $M_0$ and trained model $M_1$ of Algorithm (b) can be formulated as

$$\Delta\,\mathrm{eRank}^{(b)}(\mathcal{D}, M_0, M_1) = \mathrm{eRank}^{(b)}(\mathcal{D}, M_0) - \mathrm{eRank}^{(b)}(\mathcal{D}, M_1).$$

To compare these two ways for defining Diff-eRank, we conduct experiments using OPT models on dolly-15k dataset. The experimental results in Figure 4 demonstrate that Diff-eRank consistently increases across model sizes, irrespective of whether Algorithm (a) or Algorithm (b) is used. This observation verifies that the increasing trend for Diff-eRank is robust across different algorithms of effective rank defined on a dataset.

### 6.3 Measure Diff-eRank on Different Layers

In our research, we predominantly concentrate on the last layer of LLMs, as it usually represents the most comprehensive information encoded by the model. This layer may offer the most indicative measure of Diff-eRank. Besides, we also extend our experiments to encompass additional layers within the models. Specifically, our investigations include analyses of the first layer, the middle layer, and the last layer for language models in the OPT [45] family on dolly-15k [8] dataset. Our findings in Table 4 reveal that only the Diff-eRank on the last layer reveals an increasing trend across model sizes, which indicates that it's reasonable to analyze data representation through the last layer that encodes the most comprehensive information of the model. This may be interpreted that LLM is an integrated system where information processing occurs across the entire architecture. If we rely on early layers for analyzing Diff-eRank, this could lead to a loss of important information and we may miss crucial information processing that occurs in subsequent layers. The last layer, on the other hand, integrates this information, providing a more complete representation of the input data. The observation in our experiments reveals that early layers do not exhibit clear patterns in terms of Diff-eRank. This underscores the importance of considering the model as a whole when analyzing the representation.

## 7 Conclusion and Discussion

We introduce Diff-eRank, a new metric that can measure the "noise reduction" ability of LLM based on data representation and reflects the extent to which a pre-trained LLM eliminates the redundant dimension in the information-theoretic sense. Our method reveals the geometric characteristics of the data and is grounded in information theory. The empirical investigations show that the Diff-eRank increases when the model scales and correlates with the trend of loss and downstream task accuracy. Moreover, we use this metric to define the alignment metrics for multi-modal LLMs and find contemporary models align very well.

However, we haven't conducted experiments to observe the change of Diff-eRank during the LLMs' whole pre-training and post training stages due to the limited computation resources. Future research may broaden the investigative scope by introducing the Diff-eRank in LLMs' complete training stages. In addition, some useful techniques like pruning, quantization, and distillation may benefit from such metrics that reveal internal redundancies. The Diff-eRank metric may aid in identifying which parts of the model can be compressed without significant loss of information. We hope that Diff-eRank will open up avenues for future studies to explore how such internal representation metrics can be integrated into different potential cases.

## Acknowledgement

This project was funded by National Natural Science Foundation of China (62406192) and MSR Asia StarTrack Scholars Program. The authors also thank Kai Chen (Beijing Academy of Artificial Intelligence) for the support of computation resources.

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

# Appendix

## A Implementation Details

### A.1 Language Datasets

**Pre-training Datasets**. All sizes of OPT models are pre-trained on various datasets, including Wikipidea [14] and openwebtext2 [15]. Due to resource constraints, we select the subset of these datasets by random sampling 10 thousand pieces of data ((which is further discussed in Section D)) for the Diff-eRank observation. In addition to the datasets utilized for pre-training the models, we also incorporate supplementary datasets that were not directly involved in the OPT model's pre-training process as follows.

**Instruction-Tuning Datasets**. For the Diff-eRank observation, we choose dolly-15k [8], which is generated by human employees, as one of the instruction datasets. Specifically, we select the "context" part of this dataset as input because it contains more informative text.

**RLHF Dataset**. We utilize hh-rlhf [2] that consists of human preference data about helpfulness and harmlessness as the RLHF dataset. Each piece of this dataset contains a pair of texts, one "chosen" and one "rejected". We feed the "chosen" part of the dataset into models and calculate the performance indicators.

**Benchmark Datasets**. For the observation of benchmark indicators, we select openbookqa [22], winogrande [28] and piqa [3] for evaluation. These benchmarks are structured in a multiple-choice format. We combine the question and correct answer of each piece of data as inputs.

### A.2 Multi-modal Model Architecture

Recent Multi-modal Large Language Models (MLLMs) utilize similar model architecture by constructing a simple connector network to align the vision encoder with the LLM. This architecture is simple and efficient in aligning the vision and language information, utilizing strong LLM as the "CPU" of the multi-modal model. We showcase the architecture of LLaVA-1.5 and MiniGPT-v2 used in our experiments in Table 5.

Table 5: The model architecture comparison between LLaVA-1.5 and MiniGPT-v2.

| ARCHITECTURE | LLAVA-1.5 | MINIGPT-v2 |
|---|---|---|
| VISION ENCODER | CLIP-VIT [26] | EVA-VIT [13] |
| CONNECTOR | MLP | LINEAR |
| LLM | VICUNA-v1.5 [48] | LLAMA-2-CHAT [39] |

### A.3 Compute Resources

We conduct our experiments using NVIDIA A800-80G GPUs. The experimental time using a single A800 for calculating the Diff-eRank for a 1.3B LLM on the dolly [8] dataset is around 1 hour.

## B Complete Experimental Results

Table 6 contains the complete results for the comparison of Diff-eRank and reduced loss based on OPT [45] family considered in Figure 1. Table 7 and Table 8 illustrate the numerical results of different model families when using Diff-eRank and reduced loss for evaluation. Table 9 showcases the whole ablation results discussed in Section 6.2.

Table 6: Language modeling indicators on dolly-15k, Wikipedia, openwebtext2 and hh-rlhf.

| DATASETS | INDICATORS | OPT MODELS SIZE | | | | |
|---|---|---|---|---|---|---|
| | | 125M | 1.3B | 2.7B | 6.7B | 13B |
| DOLLY-15K | DIFF-ERANK (↑) | 54.35 | 76.39 | 83.02 | 89.60 | 89.81 |
| | $\Delta L$ (↑) | 7.6838 | 8.322 | 8.4471 | 8.5961 | 8.6505 |
| WIKIPEDIA | DIFF-ERANK (↑) | 83.55 | 136.20 | 148.59 | 161.09 | 162.88 |
| | $\Delta L$ (↑) | 7.8671 | 8.4575 | 8.5746 | 8.7009 | 8.7581 |
| OPENWEBTEXT2 | DIFF-ERANK (↑) | 103.23 | 184.76 | 205.48 | 254.30 | 262.70 |
| | $\Delta L$ (↑) | 7.8090 | 8.3601 | 8.4697 | 8.5915 | 8.6396 |
| HH-RLHF | DIFF-ERANK (↑) | 53.02 | 76.44 | 82.82 | 90.41 | 93.30 |
| | $\Delta L$ (↑) | 8.1041 | 8.4800 | 8.5242 | 8.5914 | 8.6928 |

Table 7: Comparison of Diff-eRank, reduced cross-entropy loss, and benchmark accuracy for models in OpenELM [21] family.

| MODEL SIZE | 270M | 450M | 1.1B | 3B |
|---|---|---|---|---|
| DIFF-ERANK (↑) | 114.76 | 115.69 | 117.53 | 119.40 |
| $\Delta L$ (↑) | 8.5164 | 8.6417 | 8.8210 | 9.0060 |
| ACC (↑) | 0.6183 | 0.6516 | 0.6865 | 0.7188 |

Table 8: Comparison of Diff-eRank, reduced cross-entropy loss, and benchmark accuracy for models in Cerebras-GPT [11] family.

| MODEL SIZE | 111M | 256M | 590M | 1.3B | 2.7B | 6.7B | 13B |
|---|---|---|---|---|---|---|---|
| DIFF-ERANK (↑) | 42.48 | 42.68 | 45.90 | 46.00 | 53.90 | 67.13 | 77.78 |
| $\Delta L$ (↑) | 7.1540 | 7.5343 | 7.7891 | 8.0733 | 8.3235 | 8.5339 | 8.5152 |
| ACC (↑) | 0.5410 | 0.5620 | 0.5625 | 0.5925 | 0.6300 | 0.6705 | 0.7060 |

Table 9: Comparison of Algorithm (a) and Algorithm (b) for models in OPT [45] family.

| MODEL SIZE | 125M | 1.3B | 2.7B | 6.7B | 13B |
|---|---|---|---|---|---|
| ALGORITHM (B) | 66.81 | 103.78 | 114.60 | 128.99 | 131.42 |
| ALGORITHM (A) | 54.35 | 76.39 | 83.02 | 89.60 | 89.81 |

Table 10: Comparison of metrics across different training stages.

| METRICS/TRAINING STAGES | RANDOM INITIALIZED | INITIALIZED FROM OPT-1.3B | FULLY TRAINED | OVERFITTING |
|---|---|---|---|---|
| DIFF-ERANK | 0.000 | 2.140 | 2.161 | 2.156 |
| LOSS | 10.830 | 4.692 | 4.654 | 4.663 |
| ACCURACY | 0.250 | 0.332 | 0.340 | 0.336 |

## C    Additional Experiments

To further investigate how "Diff-eRank" changes during training, we conduct additional experiments to observe the behavior of Diff-eRank across different training stages for a fixed model size. In particular, we fix the model size by using the pre-trained OPT-1.3B [45] model and continually train it on a cleaned Wikipedia [14] dataset.

According to the additional experimental results in Table 10, we observe that the trend of Diff-eRank, first increasing before fully trained and then slightly decreasing when overfitting, aligns well with the trend of benchmark accuracy and the opposite trend of loss. This suggests that Diff-eRank may serve as a complementary metric that helps understand the LLM's "noise reduction" behavior during training, and monitor the training progress.

## D    Additional Ablation Study

As mentioned in Appendix A.1, random sampling is employed to extract subsets from the whole datasets of Wikipedia [14] and openwebtext2 [15], each subset comprising 10,000 data entries, as these pre-training datasets are too large for computation. To assess the robustness of Diff-eRank in random selection, we incorporate variations in the sample sizes of the Wikipedia dataset in this ablation study. Table 11 illustrates that fluctuations in the sample size bring insignificant influence on the Diff-eRank, which affirms the stability of Diff-eRank in random sampling. Thus, this ablation study indicates the rationality of the random sampling process when dealing with large pre-training datasets in our experiments.

Table 11: Ablation study of different sampling strategies on the Wikipedia [14] dataset.

| MODEL | SAMPLING STRATEGY | | | STANDARD DEVIATION |
|---|---|---|---|---|
| | 10000 | 5000 | 1000 | |
| OPT-1.3B | 136.20 | 132.39 | 136.14 | 1.782 |

