# OpenReview forum: "Diff-eRank: A Novel Rank-Based Metric for Evaluating Large Language Models"
_NeurIPS.cc/2024/Conference — NeurIPS 2024 poster_

### Official Review · Reviewer_LQXh · 2024-06-18

**Soundness:** 3
**Presentation:** 3
**Contribution:** 3
**Rating:** 6
**Confidence:** 4

**Summary:**

This paper explores the potential of using the rank of model hidden states for evaluating model capabilities. Specifically, the authors propose calculating the rank difference between trained and untrained models as a measure of model performance. The core idea is that the rank difference can reflect the model's extent of "noise reduction". The authors compared the consistency of rank difference with other metrics (loss, accuracy) under multiple model series settings. The results showed the potential of rank difference in assessing model capabilities within certain model series and measuring the cross-modal alignment of MLLMs.

**Strengths:**

- To the best of my knowledge, this paper is the first to explore and discuss the idea of using the rank of hidden states to evaluate model capabilities, which sounds promising.

- The paper provides an intuitive understanding and theoretical significance of using rank difference as an evaluation metric.

- The experiments in this paper are relatively comprehensive, and the writing is quite clear.

**Weaknesses:**

- My main concern lies in the fact that for models within a series, the dimensionality of their hidden states is often positively correlated with size, which also implies a positive correlation with performance. A larger dimensionality of hidden states usually means a larger rank difference when the effective rank proportion remains the same. In other words, although larger models have a higher rank difference, the proportion of noise reduction might be smaller compared to smaller models. Does this suggest that rank difference may not accurately measure the extent of noise reduction? Could rank ratio be a more reasonable metric (erank_M1/erank_M0 in eq.3)?

- Although the authors observed a positive correlation between rank difference and model size within a single model series, this conclusion no longer holds when comparing models from different series simultaneously, as shown in Figure 3. Is this due to the varying dimensionality of hidden states across different models (where rank ratio might be effective)? Or is it caused by the different training methods employed for various models? If rank difference cannot be used to compare models from different series, what advantages does it offer compared to traditional metrics such as accuracy?

- When the number of test set samples lies between the dimensionality of hidden states for the large and small models being evaluated, Q in Equation 1 will take values from N and d, respectively. Could this lead to non-robust results? I believe experiments should be conducted to investigate this scenario.

**Questions:**

See Weaknesses.

I am looking forward to authors' responses and open to rise my score.

**Limitations:**

The main limitation of this paper is that it fails to specifically discuss the advantages of rank difference compared to other metrics, especially considering that the rank differences of models from different series are not on the same scale.

---

> ### Author Rebuttal · Authors · 2024-08-07
>
> Q1: A larger dimensionality of hidden states usually means a larger rank difference when the effective rank proportion remains the same. In other words, although larger models have a higher rank difference, the proportion of noise reduction might be smaller compared to smaller models. Does this suggest that rank difference may not accurately measure the extent of noise reduction? Could rank ratio be a more reasonable metric (erank_M1/erank_M0 in eq.3)?
>
> A1: Thank you for your thoughtful feedback and for suggesting the rank ratio as an alternative metric.  Below, we'd like to address this concern and explain our rationale for using rank difference over rank ratio.
>
> Firstly, we have also considered this metric initially and we agree that both rank difference and rank ratio can provide valuable insights. Each metric captures a different aspect of the change in effective rank between untrained and trained models. **The rank difference metric is designed to capture the absolute reduction in uncertainty or noise, which reflects the model's ability to compress and structure information.** This can be interpreted as a greater capacity for the model to distill relevant information and discard noise, even if the proportional change (as measured by rank ratio) might be smaller.
>
> Secondly, a decrease in rank ratio ($\frac{erank_{M_1}}{erank_{M_0}} = 1 - \frac{erank_{M_0} - erank_{M_1}}{erank_{M_0}}$) as model size grows ($erank_{M_0}$ will also grow) typically corresponds to an increase in rank difference ($erank_{M_0} - erank_{M_1}$). However, the converse is not necessarily true. Our additional experiments in Table A below reveal that the rank ratio exhibits an oscillating downward trend as model size increases, indicating a degree of instability. In contrast, we found that rank difference demonstrates a more consistent and stable pattern of increase with model size. **This inconsistency suggests that the rank ratio may not reliably capture the extent of noise reduction or the model's ability to structure information.**
>
> | Datasets/Models | OPT-125m | OPT-1.3B | OPT-2.7B | OPT-6.7B | OPT-13B |
> | --------------- | -------- | -------- | -------- | -------- | ------- |
> | Dolly  | 0.5398| 0.4719| 0.4548 | 0.4485 | 0.4593 |
> | Wiki |0.4761 | 0.3870| 0.3717| 0.3667|0.3822|
> | Openwenbtext2 | 0.4984|0.3936 |0.3761 | 0.3443|  0.3555 |
> | hh-rlhf  |0.5180 |  0.4396| 0.4218|0.4062|  0.4242|
>
> Table A: Using rank ratio as a metric for evaluation
>
> In summary, while the rank ratio provides an interesting perspective, our empirical experiments support the use of rank difference as a more reliable metric for evaluating noise reduction in LLMs.
>
> ---
>
> Q2: Although the authors observed a positive correlation between rank difference and model size within a single model series, this conclusion no longer holds when comparing models from different series simultaneously, as shown in Figure 3. Is this due to the varying dimensionality of hidden states across different models (where rank ratio might be effective)? Or is it caused by the different training methods employed for various models? If rank difference cannot be used to compare models from different series, what advantages does it offer compared to traditional metrics such as accuracy?
>
> A2: Thank you for your question. We would like to clarify this point. It is indeed normal that rank difference cannot be directly compared across different model families. **This is primarily due to the varying architectures, training methodologies, and objectives used in different model series.**
>
> **Besides, we would like to emphasize that during the LLM's pretraining phase, the focus is primarily on the validation loss rather than accuracy.** In addition, we have conducted further experiments that demonstrate that lower loss across different model families does not necessarily correlate with better performance. For instance, in our tests on the Wiki dataset, we observe that the test loss of LLaMA-7b is 1.72, while the LLaMA2-7b is 1.66 and the LLaMA3-8b is 1.85. While LLaMA3-8b has the highest loss, it actually demonstrates the best overall performance. **This example clearly illustrates that loss values cannot be reliably used to compare performance across different model families.**
>
> Given these observations, the proposed rank difference, while not suitable for cross-family comparisons, still offers valuable insights within a single model series. **It shifts the evaluation towards model representations and provides new insights into the theoretical understanding of LLM's behavior, which traditional metrics like accuracy or loss alone may not capture.**
>
> ---
>
> Q3: When the number of test set samples lies between the dimensionality of hidden states for the large and small models being evaluated, Q in Equation 1 will take values from N and d, respectively. Could this lead to non-robust results? I believe experiments should be conducted to investigate this scenario.
>
> A3: Thanks for your question. In fact, this situation won't lead to non-robust results in our evaluations. Specifically, in our experiments, we did not encounter a scenario where d is consistently smaller than N, primarily because N would exceed the maximum input context length of the models. However, we have conducted experiments where d is consistently larger than N. Specifically, in Section 4.3, we select datasets where the sequence length N for each sample is less than 100, which is smaller than the dimensionality d of the models, as even the smallest model has a dimensionality of 768. Our findings in Section 4.3 indicate that the rank difference still increases as the model scales up. Thus, the results are robust even when d is consistently larger than N. **This consistency suggests that though d is sometimes larger than N and N is sometimes larger than d, it does not lead to non-robust results.**

---

> > ### Comment · Reviewer_LQXh · 2024-08-13
> >
> > Sorry for the late. Thank you very much for the detailed responses for addressing my concerns. However, I still have some questions:
> >
> > Considering that rank difference = erank_M1 - erank_M0, both erank_M1 and erank_M0 will increase as the hidden state dimensions grow. Rank difference can only remain unchanged if erank_M1 and erank_M0 increase by the same scale, which implies that erank_M0 needs to increase by a larger proportion (given erank_M0 < erank_M1), which is evidently challenging. Therefore, an increase in rank difference with larger hidden state dimensions is a natural outcome. Given that model size is positively correlated with the model's hidden state dimensions, it is also quite natural for there to be a positive correlation between model size and rank difference.
> >
> > I believe a potential way to address my concerns is to fix the model size and compare the rank differences of different checkpoints. Ideally, the checkpoints should include cases of under-training, fully training (achieving the best accuracy), and overfitting. If a trend of rank difference first increasing and then decreasing is observed (with the best performance at the fully trained checkpoint), then the conclusions of the paper would be sufficiently reliable. Otherwise, if accuracy shows a trend of first increasing and then decreasing, while rank difference does not, then rank difference may not be a suitable substitute for accuracy.

---

> ### Author Response · Authors · 2024-08-14
> **Response to Reviewer LQXh**
>
> Thank you for your detailed and insightful feedback. We believe that the rank difference metric provides valuable insights into the internal representations and redundancy within models, which are not directly captured by traditional metrics like accuracy or loss. This makes rank difference a useful complementary metric for understanding model behavior.
>
> We also understand your concern regarding the natural correlation between model size and rank difference, and we agree that your suggested experiment, comparing rank differences across different training checkpoints for a fixed model size, would provide a more robust validation of our metric.
>
> Following your suggestions, **we have conducted additional experiments to observe the behavior of rank difference across different training stages for a fixed model size**. In particular, we fix the model size by using the pre-trained OPT-1.3B model and continually train it on a cleaned Wikipedia dataset. We evaluate checkpoints at various stages of training, including random initialized (untrained), initialized from OPT-1.3B (pre-trained OPT-1.3B), fully trained (achieving the best accuracy on Openbookqa benchmark), and overfitting. The results of rank difference, loss, and accuracy are presented in the table below.
>
>
> | Metrics/Training Stages |Random Initialized|Initialized from OPT-1.3B|Fully Trained|Overfitting
> | ------- | ---- | ---- | ---- | ---- |
> | Rank Difference| 0| 2.140| 2.161 | 2.156
> | Loss| 10.830| 4.692| 4.654 | 4.663
> | Accuracy|0.250| 0.332 |0.340| 0.336
>
>
> According to the experimental results, **we observe that the trend of rank difference, first increasing before fully trained and then slightly decreasing when overfitting, aligns well with the trend of benchmark accuracy and the opposite trend of loss**. This suggests that rank difference can serve as a complementary metric that helps understand the model's behavior, and monitor the training progress.
>
> We hope this additional analysis addresses your concerns. Thank you again for your constructive and thoughtful suggestion. We believe our findings strengthen the reliability of our conclusions.

---

> > ### Comment · Reviewer_LQXh · 2024-08-14
> >
> > We thank the authors for there efforts to address my concerns and my concerns have mostly been solved. I update my rating accordingly.

---

> ### Author Response · Authors · 2024-08-14
> **Thanks for raising your score.**
>
> Thank you for taking the time to reassess our paper and raising the score from 5 to 6. We are grateful for your thoughtful and constructive suggestions. Your insightful feedback, along with that of the other reviewers, will be incorporated into our revised revision.
>
> Thank you again for your time and effort.

---

### Official Review · Reviewer_c8rT · 2024-07-06

**Soundness:** 3
**Presentation:** 3
**Contribution:** 3
**Rating:** 5
**Confidence:** 3

**Summary:**

This paper introduces “rank difference” that measures the reduction in the rank of LLM’s representations. It evaluates the quality of LLMs, which could be used in addition to the reduction in the cross-entropy loss. The idea is based on the assumption that LLM’s representations (e.g., the hidden states of each token before the classification head) encodes the semantic and syntactic information about the input sentence. Before training, these representations are expected to be chaotic (hence high rank), but after training and alignment, the representations are expected to be more structured.

Based on this idea, “rank difference” is defined as the difference between $erank_{m0}(\Sigma_S)$ and $erank_{m1}(\Sigma_S)$ where $\Sigma_S$ is the covariance matrix of representations and $erank$ is the effective rank. The difference is computed between the representation before and after training. This measures the extend to which the model can compress the information, while other metrics such as cross-entropy measure the quality of model’s prediction.

Experiments are conducted on OPT models of sizes from 125m to 13B parameters. The results show that there is an upward trend between “rank difference” and model size (which reflects model performance) on a range of datasets, similar to the reduction in cross-entropy. In addition, rank difference is applied to measure the alignment between visual modality and text modality in Llava-1.5 and MiniGPT-v2. An ablation study on different model families is shown.

**Strengths:**

This paper presents a new evaluation metric, grounded on information theory, and demonstrates that it correlates well with model performance (as measured by model size and downstream task performance). It also allows measuring the alignment between different modalities.

**Weaknesses:**

1. Although the idea is novel, its practical usability is limited. To evaluate the quality of systems (e.g., LLM, ASR, etc), downstream metrics such as accuracy, ROUGE, BLEU, WER, etc. are used as they better align with real use cases. While, a metric like cross-entropy is used as it is differentiable, allowing it to be a training loss. I’m not sure if “rank difference” will be adopted as it’s neither related to real use case nor applicable as a training objective.

2. Regarding experiments, current results (both text-only and visual-text) only show the trend post-training, so it would be more indicative of demonstrating how “rank difference” changes during training (e.g., similar to cross-entropy loss which usually decreases monotonically during training).

**Questions:**

1. How do you think this method could be adopted?
2. What is the computational cost of computing rank difference?
3. Would it be comparable across models of different representation sizes?
4. Does/how rank difference during the alignment stage (e.g., SFT or PPO/DPO) ?

**Limitations:**

Limitations section is provided, and it has covered the main points

---

> ### Author Rebuttal · Authors · 2024-08-07
>
> Q1: Although the idea is novel, its practical usability is limited. To evaluate the quality of systems (e.g., LLM, ASR, etc), downstream metrics such as accuracy, ROUGE, BLEU, WER, etc. are used as they better align with real use cases. While, a metric like cross-entropy is used as it is differentiable, allowing it to be a training loss. I’m not sure if “rank difference” will be adopted as it’s neither related to real use case nor applicable as a training objective. And, How do you think this method could be adopted?
>
> A1: Thanks for your thoughtful feedback regarding the practical usability of our proposed "rank difference" metric. Your points about the importance of downstream metrics and differentiable training losses are well-taken. We would like to address your concerns and highlight the potential value of our approach:
>
> Complementary to Existing Metrics: We agree that downstream metrics like accuracy, ROUGE, BLEU, and WER are crucial for evaluating system performance in real-world scenarios. **Our rank difference metric is not intended to replace these measures but to complement them by providing additional insights into model behavior during training and evaluation.**
>
> Insights into Model Representation: Rank difference can measure the "noise reduction" ability of LLM based on data representation and reflects the extent to which a pre-trained LLM eliminates the redundant dimension in the information-theoretic sense. **The rank difference metric provides unique insights into the internal representations of language models, which may not be directly captured by task-specific metrics.** This can be valuable for understanding model behavior from a theoretical perspective.
>
> Future Research Directions: Rank difference may open up avenues for future research to explore how internal representation metrics can be integrated into different cases. Techniques such as pruning, quantization, and distillation may benefit from metrics that reveal internal redundancies. The rank difference metric may aid in identifying which parts of the model can be compressed without significant loss of information.
>
> The adoption of new metrics in the field may take time and require extensive validation. Our work aims to contribute to the ongoing discussion about how to evaluate and understand LLMs from a new perspective. **We believe that a diverse set of evaluation tools, including both task-specific metrics and intrinsic measures like rank difference, can provide a more comprehensive view of model quality and behavior.**
>
> ---
>
> Q2: Regarding experiments, current results (both text-only and visual-text) only show the trend post-training, so it would be more indicative of demonstrating how “rank difference” changes during training (e.g., similar to cross-entropy loss which usually decreases monotonically during training). Does/how rank difference during the alignment stage (e.g., SFT or PPO/DPO)?
>
> A2:
> Thanks for your suggestion. In our paper, we haven’t conducted experiments to observe the change of rank difference during training due to the limited computation resources at first. To address your concern and further investigate how "rank difference" changes during training, we conduct additional experiments by continually training OPT-1.3B on cleaned wikipedia dataset. We present the results in the table below.
>
> |Metrics/Training Stages|1|2|3|4|5|6
> |-|-|-|-|-|-|-|
> |rank difference ($\uparrow$)|2.148|2.154|2.158|2.160|**2.161**|2.156
> |loss ($\downarrow$)|4.655|4.654|4.653|**4.653**|4.654|4.663
> |acc ($\uparrow$)|0.332|0.334|0.336|0.332|**0.340**|0.336
>
> The additional results indicate that the rank difference shows a gradual increase during the model's training. The rank difference converges together with accuracy, which is later than the fluctuation of loss. It suggests that rank difference may be a useful metric for monitoring the training progress as it better correlates with the trend of accuracy.
>
> ---
>
> Q3: What is the computational cost of computing rank difference?
>
> A3: Thank you for your question. Below, we provide a detailed breakdown of the steps to reveal the computational cost.
>
> Normalization of Representations: For a set of representations, the complexity is $O(Nd)$ for mean subtraction and $O(Nd)$ for normalization, resulting in a total complexity of $O(Nd)$.
>
> Construction of Covariance Matrix: The covariance matrix computation involves matrix multiplication, which has a complexity of $O(Nd^2)$.
>
> Obtaining Singular Values: For a $d \times d$ matrix, the complexity of SVD is $O(d^3)$ to obtain singular values.
>
> Calculation Effective Rank: The effective rank calculation involves summing over $d$ singular values, which has a complexity of $O(d)$.
>
> Combining the complexities of the individual steps, the total computational complexity for computing the rank difference is: $O(Nd) + O(Nd^2) + O(d^3) + O(d) = O(Nd^2 + d^3)$. The computational cost of computing the rank difference is primarily influenced by the dimensionality of the representations ($d$) and the number of tokens ($N$) in the dataset. Note that the relationship between $N$ and $d$ can vary depending on the model size. In smaller models, $N$ may be larger than $d$. In larger models, $N$ may be smaller than $d$.
>
> ---
>
> Q4: Would it be comparable across models of different representation sizes?
>
> A4: Thanks for your question. We'd like to clarify that our experimental design specifically addresses this concern. **Our study is actually designed to measure models within the same family of different (representation) sizes in Section 4.** By focusing on the same model family, we ensure that the fundamental architecture and training paradigm remain consistent, while the model sizes and representation sizes vary. We believe this approach provides a fair and meaningful comparison, as it isolates the effect of model size and representation dimensionality while controlling for other variables.

---

### Official Review · Reviewer_YACX · 2024-07-10

**Soundness:** 2
**Presentation:** 2
**Contribution:** 3
**Rating:** 5
**Confidence:** 4

**Summary:**

The article presents a measure known as "rank difference" to assess the effectiveness of Language Models (LLMs) by analyzing their internal representations. This metric is based on information theory and geometric principles aiming to quantify how LLMs eliminate unnecessary information post training. The authors illustrate the utility of rank difference, in both modal (language) and multi modal contexts.

In terms of language models the study reveals that the rank difference grows with model size indicating noise reduction capability. This pattern aligns with metrics such as entropy loss and accuracy. Regarding modal models the authors suggest an assessment approach utilizing rank difference to evaluate alignment quality across modalities showing that contemporary multi modal LLMs demonstrate strong alignment performance.

Key contributions of the paper include;

Introducing rank difference as a metric for gauging the "noise reduction" capability of trained language models.
Demonstrating the correlation between rank difference, loss metrics and downstream task accuracy underscoring its potential as an evaluation criterion.
Defining alignment measures, between linguistic modalities showcasing that modern multi modal LLMs excel in achieving alignment.

The study provides real world data that backs the idea of utilizing rank variance, in datasets and model scales indicating its reliability and effectiveness in assessing language learning models. In general the research introduces a viewpoint on evaluating language learning models moving away from metrics based on predictions, to focusing on model representations. This shift brings forth perspectives on comprehending how language learning models behave.

**Strengths:**

Novelty:
The introduction of the rank difference metric offers an approach, to assessing LLMs by focusing on their representations rather than just their outputs.
Rooted in information theory and geometric principles providing a perspective on comprehending LLM behavior.
Expanding the metrics scope to modal setups and evaluating alignment coherence across different modalities.

Quality:
Assessments spanning datasets and model dimensions.
Convincing data showcasing the relationship between rank difference and traditional metrics like entropy loss and accuracy.
Analysis elucidating the mathematical underpinnings and implications of the rank difference metric.

Clarity:
Organized and easily understandable elucidation of ideas.
Thorough explanation of the proposed metric, its basis and practical implications.
Follows a sequence from problem statement to methodology and outcomes.

Significance:
Tackles queries regarding LLM evaluation underscoring the necessity for metrics that go beyond mere model results assessment.
Potential to shape how researchers and professionals evaluate and interpret LLMs offering insights, into model behavior and effectiveness.
This can be used in both mode and mode models making it versatile, for different situations.

**Weaknesses:**

The experiments have a scope;
Regarding Training Dynamics; The paper fails to delve into how the rank difference evolves throughout training missing insights, into its behavior as the model progresses.
Considering Model Diversity; Broadening the evaluation to include a range of model families and architectures would enhance the validation of the rank difference metric.

In terms of Presentation;
Clarity; Some sections require explanations to improve readability.
Visual Support; Incorporating diagrams or visual aids could boost understanding and reader engagement.

Delving Into Analysis Depth;
Insights by Layer; Analyzing rank differences across model layers would offer a holistic view.
Comparative Examination; Providing in depth comparisons with established metrics and recent studies would spotlight both strengths and limitations of the metric.

Application, in Real Scenarios;
Real world Contexts; Integrating real world application scenarios or case studies would enhance the relevance of the paper.

Computational Considerations;
Efficiency Concerns; Addressing the efficiency of calculating rank difference for large scale models and suggesting optimizations could be advantageous.

**Questions:**

Do you have any insights or early findings regarding the variations, in the rank difference metric as LLMs undergo training? Examining this aspect could offer insights into how the behaves and its usefulness throughout the model development process.

Have you thought about assessing the rank difference metric across model families or architectures aside from the OPT family? Incorporating a range of models could help validate how broadly applicable the metric is.

Could you delve deeper into comparing the rank difference metric with established evaluation metrics and recent research findings? This, in depth analysis would shed light on both the strengths and potential limitations of using the rank difference metric.

**Limitations:**

The authors recognize the limitations in conducting experiments to observe changes, in rank difference during training due to constraints. They emphasize the importance of conducting experiments to assess how applicable the metric is across model layers and its efficiency in computations.

Suggestions for Enhancements;
Training Progress; Conduct a small scale study or simulation to observe how rank difference evolves during training with models or subsets of data.
Model Variation; Assess the effectiveness of the rank difference metric across a range of model families and structures. Provide initial findings if a comprehensive evaluation is not feasible.
Computational Efficacy; Explore optimizations for calculating rank difference, such, as enhancements or parallelization methods and briefly touch upon any societal implications and ethical considerations to demonstrate awareness of broader impacts.

---

> ### Author Rebuttal · Authors · 2024-08-07
>
> Q1: The paper fails to delve into how the rank difference evolves throughout training missing insights, into its behavior as the model progresses.
>
> A1: Thanks for your question. To address your concern and further investigate how "rank difference" changes during training, we continually train OPT-1.3B on cleaned wikipedia dataset. We present the results in the table below.
>
> |Metrics/Training Stages|1|2|3|4|5|6
> |-|-|-|-|-|-|-|
> |rank difference ($\uparrow$)|2.148|2.154|2.158|2.160|**2.161**|2.156
> |loss ($\downarrow$)|4.655|4.654|4.653|**4.653**|4.654|4.663
> |acc ($\uparrow$)|0.332|0.334|0.336|0.332|**0.340**|0.336
>
> The additional results indicate that the rank difference shows a gradual increase during the model's training. The rank difference converges together with accuracy, which is later than the fluctuation of loss. It suggests that rank difference may be a useful metric for monitoring the training progress as it better correlates with the trend of accuracy.
>
> ---
>
> Q2: Broadening the evaluation to include a range of model families and architectures would enhance the validation of the rank difference metric.
>
> A2: Thank you for your comments. **We would like to clarify that we have indeed addressed this aspect in our study, specifically in Section 6.1 of our paper.**
>
> **In addition to the OPT family, we have extended our experiments to include two other model families: Cerebras-GPT and OpenELM.** These families represent a diverse range of well-trained LLMs of various sizes. **By including these different model families, we inherently incorporated a variety of architectures into our evaluation.** Each family has its unique architecture, allowing us to test the robustness of our rank difference metric across different model designs.  **As illustrated in Figure 5 of our paper, we observe a consistent increase in rank difference as models scale up across all three families.** This trend correlates with reduced loss and improved benchmark accuracy, supporting the validity of rank difference as an evaluation metric across diverse model architectures.
>
> ---
>
> Q3: Some sections require explanations to improve readability. Visual Support; Incorporating diagrams or visual aids could boost understanding and reader engagement.
>
> A3: Thanks for your suggestion. We will add more discussions shown in our rebuttal to our paper.
>
> ---
>
> Q4: Analyzing rank differences across model layers would offer a holistic view. Comparative Examination; Providing in depth comparisons with established metrics and recent studies would spotlight both strengths and limitations of the metric.
>
> A4: Thanks for your suggestions. We would like to clarify some aspects of our study to address your points:
>
> Insights by Layer: Our findings, as shown in Table 4, reveal that only the last layer demonstrates a consistent increasing trend in rank difference across model sizes. This may be interpreted as LLM is an integrated system where information processing occurs across the entire architecture. **If we rely on early layers for analyzing the rank difference, this could lead to a loss of important information and we may miss crucial information processing that occurs in subsequent layers. The last layer, on the other hand, integrates this information, providing a more complete representation of the input data.** Our experimental results indeed show that early layers do not exhibit clear patterns in terms of the rank difference. This observation supports our focus on the last layer and provides insights into how information is processed through the model.
>
> Comparative Examination: **In our study, we have indeed conducted comprehensive comparisons with the primary metrics used in LLM's evaluation, including loss and downstream task performance.** In particular, we demonstrate the correlation among our rank difference, the reduced loss and performance on downstream tasks in Section 4.2 and 4.3. This comparison shows how our metric aligns with the primary metrics. **To the best of our knowledge, loss and downstream task performance are the primary quantitative measures used for assessing LLM.** If you are aware of other specific metrics for comparison, we would greatly appreciate if you could provide references.
>
> ---
>
> Q5: Integrating real world application scenarios or case studies would enhance the relevance of the paper.
>
> A5: Thanks for your suggestion. We aim to integrate real world applications into our future work. Rank difference may open up avenues for future research to explore how internal representation metrics can be integrated into different cases. Techniques such as pruning, quantization, and distillation may benefit from this metric that reveals internal redundancies. **The rank difference metric may also aid in monitoring the LLM's training process as discussed in A2.**
>
> ---
>
> Q6: Addressing the efficiency of calculating rank difference for large scale models and suggesting optimizations could be advantageous.
>
> A6: Thank you for your suggestion. Our rank difference calculation method is designed with efficiency in mind, particularly for large-scale models. Here are some key points regarding the efficiency of our approach:
> 1. The efficiency of calculating rank difference for large scale models:
>    - We use PyTorch's efficient tensor operations, which are optimized for GPU computation. **The core operations (normalization, covariance calculation, and effective rank computation) are matrix operations** that scale well with increasing model sizes.
>    - **The rank difference can be calculated using a relatively small dataset**, further reducing computational requirements.
> 2. Potential Optimizations:
>     For extremely large models, we could implement batch processing of the representations to reduce memory requirements.
>
> We are actively exploring additional techniques to enhance the efficiency of our method in the future. We appreciate your attention to this important aspect of our work.

---

### Official Review · Reviewer_y598 · 2024-07-13

**Soundness:** 2
**Presentation:** 2
**Contribution:** 2
**Rating:** 5
**Confidence:** 2

**Summary:**

The paper proposes a rank-based evaluation metric that quantifies the amount of redundant information in the hidden representations of a model and applies it to both text-only and multi-modal models. The effective rank is obtained by the rank of its covariance matrix and is interpreted using information theory. This rank represents the degree of variations among the principal components in data.  The rank difference between two models is used to measure how much redundant information is reduced in a model relative to another. The proposed metric aligns well with commonly used evaluation metrics (loss and accuracy).

**Strengths:**

1. The paper presents a new evaluation metric through the view of the geometric property of representation matrices and information theory and shows that it aligns well with commonly used metrics like loss and accuracy for LLMs.
2. Experiments show how the rank difference varies by length, choice of the layer from which representations are extracted, type of models (text vs multi-modal), and algorithm design and can potentially guide future work on model compression.

**Weaknesses:**

1. It is unclear what additional information the absolute rank or the rank difference brings to the table apart from a new interpretation. The rank differences are hard to interpret (L261: both models align well with a high alignment score) given a lack of detail on how they are computed for models of varying sizes.
2. For the multi-modal models, the authors again propose two metrics: image reduction ratio and image-text alignment and while the reduction ratio is a bit intuitive, it is unclear why image-text alignment is defined as is and why we need the different ranks (erank 3, erank 4, and erank5). In Table 2, the reduction and alignment follow opposite trends, is it something informative?

**Questions:**

1. L224-225: can you provide any explanation on why these occasional outliers appear? This is much vivid in Table 4 where any other layer does not follow this upward trend.
2. Was there any experiment also on how the rank difference involves over the training of a model and whether that signal is informative on how long a model can be trained?
3. Is there any explanation on why the different layers are not informative of the representation redundancy (table 4)?

**Limitations:**

yes

---

> ### Author Rebuttal · Authors · 2024-08-07
>
> Q1: It is unclear what additional information the absolute rank or the rank difference brings to the table apart from a new interpretation. The rank differences are hard to interpret given a lack of detail on how they are computed for models of varying sizes.
>
> A1:
> Thanks for your comments. We would like to provide further explanations as follows.
>
> Firstly, as we have discussed in Section 1, **the effective rank of the hidden representations extracted by a model from a dataset can be considered as the uncertainty or randomness from (quantum) information theory. Meanwhile, the rank difference quantifies the reduction in uncertainty or noise in the representations before and after training for the model.**
>
> Secondly, we interpret that **as training progresses, a model's representations transition from being random and high-dimensional to more structured and lower-dimensional. The rank difference reflects this transition by measuring the extent to which redundant information has been reduced**, which is also discussed in Section 1 and 3.
>
> Thirdly, the rank difference is computed based on the effective rank of the covariance matrix of the data representations (shown in Section 3), and it does not rely on model sizes. **Rank difference can be directly calculated and compared among models of varying sizes within the same family.**
>
> Q2: For the multi-modal models, it is unclear why image-text alignment is defined as is and why we need the different ranks. The reduction and alignment follow opposite trends. Is it something informative?
>
> A2:
> Thanks for your questions. We would like to highlight these key aspects:
>
> Firstly, we want to design a metric to evaluate the degree of alignment between the image and text modalities, ranging from 0 to 1. When the MLLM reaches the perfect alignment, the metric is close to 1. As the alignment worsens, the metric will decrease. **In particular, the absolute rank can be seen as the amount of absolute uncertainty or randomness.** The ranks for individual images (erank3), text (erank4), and image-text pairs (erank5) show how much the model integrates and represents each modality. **If these three ranks from different modalities are close to each other, it means that they align well from the perspective of information theory.** Thus, we design this metric to reflect the degree of closeness (i.e., alignment) among different modalities.
>
> Secondly, regarding the observation where Image Reduction Ratio and Image-Text Alignment follow opposite trends, **it is important to note that these two metrics measure different aspects of the MLLMs and should be considered independently.** In particular, Image Reduction Ratio focuses on the model’s ability to condense visual information, while Image-Text Alignment measures the quality of alignment between different modalities. **The trends observed in Table 2 do not imply a direct relationship between the two metrics. Instead, they highlight different dimensions of MLLM's performance.**
>
> Q3: Can you provide any explanation on why these occasional outliers appear?
>
> A3: Thanks for your question. Actually, the observed deviations are within an acceptable range and do not affect the robustness of our findings. In particular, several factors may contribute to the presence of these outliers:
>
> Training randomness: The training process of large language models inherently involves randomness. **Even within the same model family, slight differences in training dynamics can affect the performance in our evaluations.**
>
> Outliers are not unexpected: It is worth noting that outliers are not unique to the rank difference. **Other commonly used metrics, such as loss and accuracy, also exhibit occasional outliers in their trends across different model sizes. For instance, OPT-6.7B gains lower performance on openbookqa than OPT-2.7B in Table 1.** Thus, outliers are not unexpected in the evaluation of LLMs with a single metric.
>
> Despite these occasional outliers, the overall trend remains consistent and supports our conclusion that rank difference generally correlates with loss reduction and accuracy improvement as model size increases.
>
> Q4: Was there any experiment also on how the rank difference involves over the training of a model and whether that signal is informative on how long a model can be trained?
>
> A4:
> Thanks for your question. To address your concern and further investigate how "rank difference" changes during training, we continually train OPT-1.3B on cleaned wikipedia dataset. We present the results in the table below.
>
> |Metrics/Training Stages|1|2|3|4|5|6
> |-|-|-|-|-|-|-|
> |rank difference ($\uparrow$)|2.148|2.154|2.158|2.160|**2.161**|2.156
> |loss ($\downarrow$)|4.655|4.654|4.653|**4.653**|4.654|4.663
> |acc ($\uparrow$)|0.332|0.334|0.336|0.332|**0.340**|0.336
>
> The additional results indicate that the rank difference shows a gradual increase during the model's training. The rank difference converges together with accuracy, which is later than the fluctuation of loss. It suggests that rank difference may be a useful metric for monitoring the training progress as it better correlates with the trend of accuracy.
>
> Q5: Is there any explanation on why the different layers are not informative of the representation redundancy?
>
> A5:  Thanks for your comment. We would like to offer several potential explanations: LLM is an integrated system where information processing occurs across the entire architecture. **If we rely on early layers for analyzing the rank difference, this could lead to a loss of important information and we may miss crucial information processing that occurs in subsequent layers.** The last layer, on the other hand, integrates this information, providing a more complete representation of the input data. We indeed observe that early layers do not exhibit clear patterns in terms of the rank difference in our experiments. This underscores the importance of considering the model as a whole when analyzing the representation.

---

> > ### Comment · Reviewer_y598 · 2024-08-12
> >
> > Thank you for the response. I have read other reviews, the author's rebuttal, and updated my scores accordingly. It would be very useful to integrate the explanations (including the outliers, the training evolution and the explanation for over layers) in the next version of the paper.

---

> ### Author Response · Authors · 2024-08-12
> **Thanks for raising your score.**
>
> Thank you for your reconsideration of our paper and the adjustment of the score. We are very grateful for your acknowledgment of the empirical contribution our work provides to the field. We assure you that the valuable suggestions and insights from you and other reviewers, as well as our explanations, will certainly be integrated into our revised version.
>
> We sincerely appreciate the time and effort you've dedicated to this. Thanks again for your review and comments.

---

### Decision · Program_Chairs · 2024-09-25

**Decision:**

Accept (poster)

**Comment:**

This paper proposes a new rank-based evaluation metric for LLMs which quantifies the amount of redundant information in the hidden representations of a model and applies it to both text-only and multi-modal models. The idea is to compute the "rank difference" between trained and untrained models and use it as a measure of model performance. Experiments show the potential of rank difference in assessing model capabilities within certain model series and measuring the cross-modal alignment of MLLMs.

Reviewers had mixed feelings about this paper. They pointed out as strengths the originality of the proposed metric and its theoretical soundness, as well as their empirical alignment with other metrics and the potential for rank difference to guide future work on model compression. They also pointed out the comprehensive experiments and the clarity of the writing. The main weaknesses are the lack of clarity about the additional information that the proposed metrics bring to the table (particularly for multi-modal models), as well as a a concern that rank difference may not accurately measure the extent of noise reduction, due to the correlation between the dimensionality of the hidden states (and model size) with performance, which makes larger models to have a higher rank difference but less "noise reduction" than smaller models.

Overall, I am leaning towards acceptance. The proposed "rank difference" is original and it seems useful not only for evaluation but potentially for model development and compression. The authors addressed in detail all the questions asked by the reviewers and in my opinion they alleviated some of their concerns. They reported additional results which seem to address the main weaknesses. I believe the missing details can be incorporated easily in the final version and enhance the paper.